# Exploring the Mycovirus Universe: Identification, Diversity, and Biotechnological Applications

**DOI:** 10.3390/jof9030361

**Published:** 2023-03-15

**Authors:** Diana Carolina Villan Larios, Brayan Maudiel Diaz Reyes, Carlos Priminho Pirovani, Leandro Lopes Loguercio, Vinícius Castro Santos, Aristóteles Góes-Neto, Paula Luize Camargos Fonseca, Eric Roberto Guimarães Rocha Aguiar

**Affiliations:** 1Department of Biological Sciences, Center for Biotechnology and Genetics, Universidade Estadual de Santa Cruz, Ilhéus 45662-900, Bahia, Brazil; carolarios.312@gmail.com (D.C.V.L.); 1994bmaudiel@gmail.com (B.M.D.R.); pirovanicp@gmail.com (C.P.P.); leandro@uesc.br (L.L.L.); 2Department of Biochemistry and Immunology, Instituto de Ciências Biológicas, Universidade Federal de Minas Gerais, Belo Horizonte 31270-901, Minas Gerais, Brazil; vini8cs@gmail.com; 3Department of Microbiology, Instituto de Ciências Biológicas, Universidade Federal de Minas Gerais, Belo Horizonte 31270-901, Minas Gerais, Brazil; arigoesneto@gmail.com; 4Department of Genetics, Instituto de Ciências Biológicas, Universidade Federal de Minas Gerais, Belo Horizonte 31270-901, Minas Gerais, Brazil

**Keywords:** mycovirus, fungi, biological control, technologies for mycovirus identification, virus-fungi interactions

## Abstract

Viruses that infect fungi are known as mycoviruses and are characterized by the lack of an extracellular phase. In recent years, the advances on nucleic acids sequencing technologies have led to a considerable increase in the number of fungi-infecting viral species described in the literature, with a special interest in assessing potential applications as fungal biocontrol agents. In the present study, we performed a comprehensive review using Scopus, Web of Science, and PubMed databases to mine mycoviruses data to explore their molecular features and their use in biotechnology. Our results showed the existence of 267 mycovirus species, of which 189 are recognized by the International Committee on Taxonomy of Viruses (ICTV). The majority of the mycoviruses identified have a dsRNA genome (38.6%), whereas the *Botourmiaviridae* (ssRNA+) alone represents 14% of all mycoviruses diversity. Regarding fungal hosts, members from the *Sclerotinicaeae* appeared as the most common species described to be infected by mycoviruses, with 16 different viral families identified so far. It is noteworthy that such results are directly associated with the high number of studies and strategies used to investigate the presence of viruses in members of the *Sclerotinicaeae* family. The knowledge about replication strategy and possible impact on fungi biology is available for only a small fraction of the mycoviruses studied, which is the main limitation for considering these elements potential targets for biotechnological applications. Altogether, our investigation allowed us to summarize the general characteristics of mycoviruses and their hosts, the consequences, and the implications of this knowledge on mycovirus–fungi interactions, providing an important source of information for future studies.

## 1. Introduction

The Fungi kingdom is a diverse group of organisms, with two to five millions species having been described, and comprises of 12 phyla: Ascomycota, Basidiomycota, Entorrhizomycota, Chytridiomycota, Mucoromycota, Microsporidia, Monoblepharidomycota, Zoopagomycota, Neocallimastigomycota, Blastocladiomycota, Aphelidiomycota, and Cryptomycota/Rozellomycota [1]. Fungi can perform an array of ecological roles, including nutrient cycling and decomposition of organic matter [2]. They can also be pathogens or parasites that affect a wide range of organisms. Furthermore, some fungi species potentially impact a variety of industries, such as food, pharmaceuticals, agricultural, gastronomy (edible mushrooms), environmental health indicators, and biological control of agricultural pests, acting as phytopathogens [2,3].

Phytopathogenic fungi are known to affect specific plant organs and reduce more than 50% of agricultural crop production [4]. Despite their negative effects on human health and the environment, the use of chemical compounds, or agrochemicals, as well as genetically engineered crops with increased resistance against pathogens (when available), has been widely applied as tools for efficient Integrated Pest Management for fungal infection control [5,6]. Biological control is an alternative method for the use of agrochemicals since it consists of the regulation of the number of individuals/propagules of a given undesired species usually taking advantage of natural enemies. Bacteria and other fungi, for example, can use a virulence factor to compete for space and nutrients, antibiosis, mycoparasitism (direct control), induction of host resistance, and/or growth promotion (indirect control) to avoid the growth of phytopathogens [7]. These factors are influenced by a variety of environmental conditions that are challenging to control in field, such as temperature, humidity, UV light, and pH [8,9,10]. Despite being the most natural alternative for pathogen control, biological control is also more complex to implement and takes a longer time to reduce the infection and damage signs than the conventional one [5,9]. However, biocontrol is considered the most natural and environmentally friendly alternative for the control of pathogens. For instance, we can mention the use of plant viruses (e.g., *Tobacco mosaic virus*) to transport antifungal proteins to host plant cells in order to boost resistance to diseases [10]. The possibility of using viruses as biocontrol agents for fungal phytopathogens has been suggested over the last few years. This strategy is based on the potential of the fungi-infecting virus (also called mycovirus) to interfere with fungal cell physiology due to the viral replication. For example, the genus *Hypovirus* can naturally infect and cause hypovirulence in *Cryphonectria parasitica*, which is responsible for the disease known as Chestnut blight, reducing the fungus growth in its host tree-plant [8].

The discovery of mycoviruses occurred approximately 60 years ago in the edible fungus *Agaricus bisporus*. Early mycovirus identification was limited to phenotypic observation of the viral particles by electron microscopy [11]. With the advancement of high throughput sequencing (HTS), further identification of mycoviruses was possible for many fungal species, leading to a substantial increase in fungi-infecting viruses described in recent years [12,13]. According to the International Committee on Virus Taxonomy (ICTV) (https://talk.ictvonline.org/taxonomy/, accessed on 20 October 2022), 29 viral families display representatives that infect fungal species: *Partitiviridae, Chrysoviridae*, *Amalgaviridae*, *Curvulaviridae*, *Hypoviridae*, *Reoviridae*, *Totiviridae*, *Endornaviridae*, *Alphaflexiviridae*, *Deltaflexiviridae*, *Barnaviridae*, *Narnaviridae*, *Pseudoviridae*, *Metaviridae*, *Gammaflexiviridae*, *Mymonaviridae*, *Phenuiviridae*, *Mitoviridae*, *Polymycoviridae*, *Quadriviridae*, *Megabirnaviridae*, *sclerobunyaviridae*, *Rhabdoviridae, Fusagraviridae*, *Botybirnavirus*, *Tymoviridae*, *Mycoaspiviridae*, and *Botourmiaviridae.* In addition, many viral species have not yet been classified at family level.

The majority of mycoviruses have RNA genomes, which can be positive- (+) or negative (-)-sense single-stranded RNA (ssRNA), or double-stranded RNA (dsRNA) [14]. There are also mycoviruses with DNA genomes, but in much lower abundance [11]. All mycovirus species lack an extracellular phase. Therefore, their transmission is limited to intracellular mechanisms [15]. The two most common transmission modes are horizontal transfer, which results from cytoplasmic attachment of hyphae of the same fungal species (anastomosis), and vertical transmission through asexual spore production [16,17]. Mycoviruses are becoming more popular as plant disease biocontrol agents because they can affect their host’s fitness in ways that are either beneficial, harmful, or neutral [5,18]. They can affect phytopathogenic fungi by causing hypovirulence, with little to no environmental impact. Taking all of this together, in the present study we aimed to perform a broad and systematic review on all mycoviruses data available so far in public databases, assessing the magnitude of their diversity, host range, and potential to affect fungal-host biology. The possible use of mycoviruses as biocontrol agents was also addressed.

## 2. Methodology

### Literature Review and Acquisition of Genomic Data

A comprehensive literature review was performed to obtain information on the identification of mycoviruses and their acceptance by the ICTV until August 2021. The collected information and data were collected through the scientific literature available in the Scopus, Web of Science (WoS), and PubMed databases, using “mycovirus” and “viruses in fungi” as keywords. The corresponding workflow of this review is shown in Appendix A. The retrieved studies were evaluated by title and abstract, considering studies on biology, taxonomy, identification, and application of those viruses. Only primary research articles were included in the accepted papers for this research. Selection on the first (titles/abstract) and second levels (full-text of abstract-selected articles) were based on the central theme of the study, which included a mycoviral species or family. Articles referring to applications of fungi or viruses in human diseases, perspectives of applications of viruses in human diseases, or as cause variables in the development of cancer and old articles published prior to the year 2000, were excluded due to the updating of biological concepts on mycoviruses research [11]. Articles that did not fulfill the inclusion criteria (Appendix A) were discarded.

After manual curation, we used public information available at National Center for Biotechnology Information (NCBI) databases to extract the genomic and taxonomic information from the viral species of each selected article and to assess its molecular features, such as genome structure and length. The filtered data was organized based upon the species, genus, and viral family taxonomic categories (Appendix A). Other taxonomic information related to hosts (species, genera, and family, NCBI accession number, genome type and size) were also considered. Overview of the data and plots were constructed using R software version 4.0.3 with ggplot2 package [19]. Sankeyplot diagram was constructed using the SankeyMATIC server (http://sankeymatic.com/build/, accessed on 15 January 2023). Mycovirus genomes’ data were used to predict ORFs (considering different genetic codes) using the EMBOSS getorf tool [20]. For each annotated genome, the size of the ORFs for each genetic code tested (Standard Code (1); Vertebrate Mitochondrial Code (2); Yeast Mitochondrial Code (3); Mold, Protozoan, and Coelenterate Mitochondrial Code and Mycoplasma/Spiroplasma Code (4); Invertebrate Mitochondrial Code (5); Ciliate, Dasycladacean and Hexamita Nuclear Code (6); Echinoderm and Flatworm Mitochondrial Code (9); Euplotid Nuclear Code (10); Bacterial and Plant Plastid Code (11); Alternative Yeast Nuclear Code (12); Ascidian Mitochondrial Code (13); Alternative Flatworm Mitochondrial Code (14); Blepharisma Nuclear Code (15); Chlorophycean Mitochondrial Code (16); Trematode Mitochondrial Code (21); Scenedesmus obliquus mitochondrial Code (22); and Thraustochytrium Mitochondrial Code (23); and the presence of conserved domains (such as RNA polymerase RNA-dependent, coat, etc.)) were evaluated using the Conserved Domain Database (CDD) [21]. The information obtained is available in Appendix A and was plotted as a heatmap using the ComplexHeatmap [22] package in R [23].

## 3. Results and Discussion

### 3.1. Data Curation

A total of 1713 papers were retrieved from the three searched databases. The first applied filter removed 520 duplicated articles. The second filter was applied to the title and abstract to select only papers related to the following subjects: identification of mycovirus species; studies on mycovirus taxonomy and biology; and investigation on their applications, which allowed the removal of 949 papers. Therefore, after rigorous manual curation, we kept 35 systematic reviews, 198 original articles focused on the identification and characterization of viral and RNA molecules in fungal cells, and 11 related to the development of mycoviruses as a biological control method (Appendix A). Hence, a total of 1469 articles were eliminated for not fulfilling the research requirements, leaving us with 244 articles accepted for further analyses.

### 3.2. Mycovirus Diversity: Fungal Hosts, Main Viral Families, Genomic Aspects and Relationship with the Host Cell

According to our data, there are a high number of viral families described as capable of infecting fungi. Our literature-mining resulted in 267 mycovirus species found in the three public databases, within which only 189 (70.78%) are currently accepted by the ICTV. The *Botourmiaviridae* was the most common one in this study, which had 38 (14%) fungi-infecting species, followed by *Mymonaviridae* (11%), and *Partitiviridae* (10%) (Figure 1). Most of the mycovirus species found (136 = 62%) have dsRNA genomes, followed by +ssRNA (80 = 30%), -ssRNA (41 = 16%), and ssDNA genome (5 = 2%) (Figure 1). Until 2010, there was no mycovirus identified as having a dsDNA genome [24], most likely because this viral group lacks an envelope and, in some cases, the capsid itself, thereby hampering its identification by microscopy and chromatography approaches [25].

Fungal viruses were found infecting five different phyla from the Fungi kingdom. The Ascomycota (224 viral species) and Basidiomycota (37) comprised the majority of mycoviruses described, since they are considered the two most well-known and studied phyla [26]. The remaining three phyla were Mucoromycota (three viral species), Monoblepharidomycota and Blastocladiomycota (one species each). Mycoviruses were described as capable of infecting 81 different fungal species. The diversity of fungal hosts and their respective infecting mycoviruses are shown in Figure 2.

Mycoviruses are as ubiquitous in nature as their hosts; nevertheless, the biological and ecological roles played by these viruses are largely unknown, since a single fungal isolate can be infected by multiple mycoviruses and yet lack symptoms [27,28]. For instance, the fungi *Sclerotinia sclerotiorum*, is targeted by viruses from at least 17 viral families, some of which were found to be in co-infection in the same individual. Such a remarkable amount of information on this system is probably due to an overall interest in finding biological agents potentially capable of controlling the growth of relevant phytopathogen species from the *Sclerotinia* genus, as they affect many important crops in Asia [12,29,30,31,32,33,34,35]. Other fungal genera, such as *Heterobasidium* and *Penicillium*, were found to be affected by nine different viral species.

Other examples in the literature are the relationships of two viruses in the fungus *Rosellinia necatrix*, in which one needs the other to replicate in the cytoplasm of the fungal cell [28], and the co-infection of totivirus and hypovirus in the fungus *Cryphonectria parasitica*, in which the replication of the former is impaired by activation of the silencing machinery by the other [36].

In Figure 2, we can observe important mycovirus-fungi interactions, such as those represented in *Botourmiaviridae, Chrysoviridae,* and *Patitiviridae*, which can infect several fungal families. We also observed in Figure 2 that the connections of the fungal family *Sclerotiniaceae* showed to cover a broad range of viral families, being the target of at least 15 distinct families, besides unclassified ones. In addition, several of the viral families are specific and infect only one group of fungi (for example, *Sclerotinia*). Most of the recent studies were performed in the genus *Sclerotinia.* In total, more than 57 viruses, representing almost 22% of all mycoviruses known to date have been described as capable of infecting elements from this genus. At least 35 mycoviral species from eight families (*Rhabdoviridae*, Unassigned *Botybirnavirus*, *Tymoviridae*, *Fusagraviridae*, *Sclerobunyaviridae*, *Mycoaspiviridae*, *Alphaflexiviridae,* and *Gammaflexiviridae*) have shown specificity to *Sclerotiniaceae* [37,38,39,40]. It should be noted that most of these mycoviruses families have not yet been accepted by the ICTV [33,37].

Fungal species can use different genetic codes rather than the standard one, which reflects their translation strategies and host amplitude [41]. Since mycoviruses use the host translational machinery to express the proteins involved on their replication cycle, we evaluated the main genetic codes used by these viruses (Figure 3). The hierarchical clustering analysis, based on the length of the large ORF and presence of conserved domains (see details on the legend of Figure 3) when tested for the 17 genetic codes possible (including standard code, mitochondrial code, among others), revealed four major groups of mycoviruses according to the code usage possibilities: group 1 (on the upper part of the cluster) included 28 viral species and was essentially defined by 12 codes combined with the ORF structure class III (ORF was a smaller size than predicted by the standard one, with the presence of conserved domain); group 2 included 86 species essentially with ORF structure class II (ORF was greater in size than predicted by the standard one, with lack of conserved domain) and 14 genetic codes (all but # 2, 22 and 23); group 3, which included 46 viral species, was contained elements with ORF structures from the classes I and III (both with conserved domain), with essentially 9–11 genetic codes associated to the former ORF and 5–6 codes to the latter; finally, the last major group contained 34 viruses, and was based on 10 genetic codes and the ORF structure class III, as well as four codes and the ORF structure class I (both with conserved domain). A strong link between the codes # 2, 22, and 23, and the ORF structure class IV, was noticed for the vast majority of viruses (Figure 3); such an association has not contributed to the grouping pattern obtained. The genetic codes # 2 (vertebrate mitochondrial), 22 (*Scenedesmus obliquus* mitochondrial), and # 23 (*Thraustochytrium* mitochondrial) were the least used among the viral genomes evaluated. Other genetic codes such as # 3 (yeast mitochondrial) and # 4 (invertebrate mitochondrial), which are specific to the mitochondrial genomes of yeast and filamentous fungi, were one of the most found, indicating the presence of viruses capable of infecting the mitochondrial organelle in fungi, also called mitoviruses (Figure 3). Interestingly, these results defy the theory that proposes the immutability of the standard genetic code to avoid lethal genetic alterations [42]. Our results demonstrate that mycoviruses can use different genetic codes and are related to the genetic code use of its host and location (in case of mitochondria-infecting viruses). However, we still do not fully understand the impact of the use of distinct genetic codes on mycoviruses fitness and/or behavior. Most of which is known is derived from the identification of viruses that infect multiple hosts from different kingdoms, such as fungi and plant or insects, showing the possibility of codon usage consistent with both organisms. Nevertheless, we can suppose that one of the main impacts of using a different genetic code is that it can allow the virus to produce proteins that are different from those of the host. This can give the virus an advantage in evading the host’s immune system and in adapting to new environments. For example, some viruses have been shown to use a different genetic code to produce proteins that are resistant to antiviral drugs. Another impact of using a different genetic code is that it can affect the efficiency of protein synthesis. Some genetic codes are more efficient than others, meaning that they can produce proteins more quickly and with fewer errors. By using a more efficient genetic code, the virus can produce its proteins more quickly and effectively, giving it a competitive advantage over the host. Finally, using a different genetic code can also affect the way that the virus interacts with the host cell. For example, some viruses have been shown to use a different genetic code to produce proteins that interact with the host cell’s machinery in different ways. This can lead to changes in the way that virus replicates and spreads within the host, as well as changes in the symptoms and severity of the resulting disease.

The majority of mycoviruses replicate in the cytoplasm of fungal cells, suggesting that they have adapted to the host’s intracellular life cycle, but have lacked the ability to enter into the host cells’ nucleus [43,44,45,46,47,48]. Mycoviruses cannot induce cell lysis in their host, and cannot spread extracellularly; however, they can be transmitted intracellularly, thus resembling a symbiont in many ways [49]. In this context, three transmission mechanisms have been described: hyphal anastomosis (horizontal transmission), through asexual spores, or from mother-to-daughter cells (vertical transmission) [36,48]. In the case of fungal anastomosis, mycovirus transfer occurs by homokaryotic transmission, in which fungal isolates are able to recognize and distinguish themselves from others [50]. Non-self-recognition between isolates/species of different mycelial origins results in programmed cell death, which is termed as heterocaryon incompatibility [50,51]. Nevertheless, virus transmission through incompatible strains or species has indeed been observed in nature. For instance, *Sclerotinia sclerotiorum mycoreovirus 4* (SsMYRV4), which is associated with hypovirulence in *S. sclerotiorum*, has demonstrated the ability to repress non-self-recognition of the fungus and facilitate co-infection through horizontal transmission of mycoviruses by somatic hyphae-incompatible groups [52].

There are two hypotheses about the origin of mycoviruses. The first one is related to the theory that they are from an ancient origin and coevolved with their hosts. The second one is that they moved from a plant cell to a fungal host. Additionally, there is evidence that Partitivirus has been shared between plants and fungi across time [13]. This second hypothesis relies on the fact that plants and their invading fungi can bidirectionally interchange many chemicals linked to fungal proliferation and/or inhibition of host defense responses [53]. Some studies have used artificial inoculation to demonstrate the compatibility of some plant viruses and viroids with fungal/yeast hosts, so that plant viruses have shown to be capable of replicating in fungal cells [46,54,55]. It is also worth noting that viruses infecting a marine fungus (*Penicillium aurantiogriseum var. viridicatum*) associated with algae have shown to reproduce in plant protoplasts [56]. In our study, we observed that most of the ICTV-accepted genera (56%) have not yet shown experimental evidence confirming their respective transmission mechanism; only 85 mycoviruses have been tested in the laboratory and in the field to assess their transmission strategy and capacity. Within this group, only 44% of the elements in genera accepted by ICTV present information regarding their transmission mode, in which 52.3% showed to have vertical transmission, whereas the remaining 47.7% are transmitted horizontally.

Since fungi can infect a vast number of plants, we also investigated the diversity of plant species affected by virus-infecting fungi. In Figure 4 and in Appendix A, fungal families affecting different plant groups of economic importance are displayed. From the 79 fungal host species reported so far, 50 are known to be phytopathogenic, with the vast majority affecting an array of plant groups (ornamental and flowering plants, fruits, vegetables, legumes, and cereals). The *Sclerotiniaceae* displays the widest range of host plants: the representative species *S. sclerotiorum* can affect herbaceous, succulent, woody and monocotyledonous plants; *S. trifoliorum* has been isolated from a vegetable type of plant; *S. minor* has been discovered in sunflower, tomato, carrot, peanut, and lettuce [57]. Since *Sclerotinaceae* can infect different crop species, the study of mycoviruses as a biological control agent is of great interest, as they may be able to operate in the reduction or mitigation of fungal growth, pathogenic effects, and consequent economical losses [57]. Phytopathogenic fungal species can cause significant losses in agricultural production all over the world [58,59]. In China, yield losses can reach 80% during severe outbreaks of *S. sclerotiorum* [60], and in the United States, estimated economic losses by this fungus can reach over U$ 200 million [61]. A previous review article has documented 40 studies that report the challenges and issues generated by fungi, as well as methods for control and reduction of agricultural losses [58,59]. Viral infections have been shown to have a negative impact on fungal growth since the 1990s. Since then, several studies describing the reduction in the fungal virulence have been published. Nowadays, mycoviruses are frequently associated with reduced virulence of fungal species upon plants, since the viruses affect mycelium growth rate, spores production, and pigment content [62,63,64]. It is also possible that mycovirus infection does not result in symptomatic or hypovirulent phenotypes, reflecting the complex interactions that occur within the host fungal species [65].

### 3.3. Cases of Mycovirus as Hypervirulent Agents with Increased Pathogenicity

Despite how many mycoviruses can act by reducing the virulence of fungal species, a number of fungi-infecting viruses can actually increase the virulence of their hosts, providing them an adaptative advantage in specific environments [66]. The *Saccharomyces cerevisiae L-A virus*, for example, has a genome that encodes different genes capable of regulating the expression and secretion of various mycotoxins in its host (*S. cerevisiae* yeast), thus enabling self-immunity. Individuals infected with these viruses are considered to have a ‘killer’ phenotype. Yeasts with the killer system can grow isolated from other microorganisms because they are immune to mycotoxins produced during infection by the virus [67].

Another example occurs in the entomopathogenic fungus *Bauveria bassiana*, which shows an increase in its virulence when infected by the *Beauveria bassiana victorivirus 1* (BbVV-1) and *B. bassiana polymycovirus 1* (BbPmV-1). This hypervirulence increases the pathogenicity of *B. bassiana* during insect infections (Mediterranean fruit fly *Ceratitis capitata*) [68]. In addition, in a study conducted in *Magnaporthe oryzae*, it was found that infection by *Totivirus* increased the production of tenuazonic acid-derived mycotoxins, which were caused by virus-induced epigenetic changes [49]. A similar case occurs with *Rosellinia necatrix*, which undergoes targeted gene silencing by infection with the *Rosellinia necatrix partitivirus 2 virus* [69]. A further example is the *Rosellinia necatrix quadivirus 1* [24] that causes significant alterations in the pigmentation and rapid growth of the *Leptosphaeria biglobosa* fungus, besides causing subtle alterations in the metabolism and defenses of the *Brassica napus* plant, which results in a fungal hypervirulent phenotype [70].

### 3.4. Cases of Mycovirus as Biological Control Agents (Hypovirulent Phenotypes)

Some studies have discussed how the interaction among mycoviruses, and their fungal hosts can induce signs of hypovirulence in the fungus. For instance, *Flammulina velutipes browning virus* can drive abnormal effects in the growth of the edible fungus *Flammulina velutipes*; other signs included brown fruiting bodies and flat morphology [71]. The mushroom bacilliform fungus virus (MBV) can affect the fungus *Agaricus bisporus*, causing abnormalities in the growing mycelium [72]. In another case, the fungus *Lentinula edodes* (‘shiitake’) is affected by the *Lentimonavirus lentinulae virus* that causes imperfect browning [73]. Based on those alterations caused by mycoviruses, it has been raised as the possibility of its use as a biocontrol agent of plant pathogenic fungi, thereby serving as an alternative to chemical control. Here, we describe some examples that highlight this possibility. For instance, an *Alphapartitivirus* member, the species *Rosellinia necatrix partitivirus 2*, has shown to cause hypovirulence in *Rhizoctonia solani*, a soil basidiomycetous fungus that infects a wide range of host crops, including vegetables, ornamentals, and tree species [74]. Species from the *Heterobasidion* genus that can be infected by a diverse group of partitiviruses can induce hypovirulence in several fungal species, including geographically distributed isolates from North America and Europe [44,45,48,75]. *Heterobasidion partitivirus 13* can cause phenotypic weakening in strains of *H. ecrustosum, H. annosum* and *H. parviporum* fungal species that cause root rot in forest plant species [75], although some studies have described that *H. annosum* shows some tolerance to *Heterobasidion partitivirus 13* infection [47]. *H. partitivirus 13* has been tested in the field as a potential biocontrol agent of *Heterobasidion* spp.; a reduction in fungal growth on trees was observed, indicating this virus could reduce fungal growth on natural substrates, and so, be used as biological control. Another example is the study conducted in *Aspergillus fumigatus*, which demonstrated the existence of many mycovirus species: *A. fumigatus partitivirus-1 (AfuPV-1, PV)*, *A. fumigatus chrysovirus (AfuCV, CV)*, *A. fumigatus tetramycovirus-1 (AfuVmycovirus-1),* and *Tetramycovirus-1 (TuVmycovirus-1)* [76]. These viral species have the ability to generate phenotypic staining by inducing silencing of specific host mRNAs that are differentially expressed as a result of viral transcription [76].

Several other examples can be mentioned. The *Fusarium oxysporum* fungal pathogen, which causes vascular wilt in carnations (*Dianthus caryophyllus*), has shown a reduction in its growth rate and the spatial distribution in plant tissues when infected by the *F. oxysporum chrysovirus 1* [77]. Viral species, such as *Botryosphaeria dothidea chrysovirus 1* [78], *Magnaporthe oryzae chrysovirus 1-A* [79], *Colletotrichum fructicola chrysovirus 1* [80] and *Agaricus bisporus virus 1* [81], can also reduce virulence and generate phenotypic changes in their respective fungal hosts. The fungal species *Alternaria alternata* infected by *Chrysovirus alternaria alternata* showed reduced mycelial growth, aerial mycelial collapse, deregulated pigmentation, and cytolysis [82]. In *Cryphonectria. parasitica*, hypovirulence-causing infection has been detected by the *Cryphonectria hypovirus 1* (CHV1), the most studied mycovirus to date [83]. In Europe, this mycovirus has been used as a biological control agent for the ‘chestnut blight,’ preventing epidemics in European chestnut forests. The application of this specific biological control method is based on treating individual trees with *CHV1*-infected strains of *C. parasitica* that are subsequently spread on untreated trees, thereby preventing further infection by virulent fungal strains [84]. *Mycoreovirus 1* can induce silencing in various genes of *C. parasitica*, such as *rdr* (RNA-dependent RNA polymerase), *dcl* (Dicer), and *agl* (Argonaute), resulting in a rearrangement of gene expression that affects the mycelial growth [85]. Additionally, another study has shown that this viral species can also silence the *agl* and *dcl* genes in *Arabidopsis thaliana*, indicating the capacity of the virus to infect two different hosts and probably alters their fitness [86]. The *Sclerotinia minor endornavirus 1 (SmEV1)*, causes hypovirulence in *S. sclerotiorum* strains, since it can reduce the pathogenicity, with the fungus presenting an altered morphology characterized by numerous irregular mycelial sectors on the colony margin and abnormal mycelial growth [87]. Another example is *Helicobasidium mompa endornavirus,* which decreases the virulence of the purple root-rot fungus *H. mompa*. Colonies infected with this virus show reduced hyphal development but effective spores production, facilitating viral dissemination [88]. *Sclerotinia sclerotimonavirus negative-strand RNA virus* also induces hypovirulence in its fungal hosts, which presents as reduced growth and loss of the ability to produce sclerotia [89]. The *Megabirnavirus* genus, which has only one species, the *Rosellinia necatrix megabirnavirus 1*, can inhibit fungal growth and lead to hypovirulence in the fungi that causes both the white root rot (*R. necatrix*) and the chestnut blight (*C. parasitica*) diseases [90]. *Cryphonectria* species are also impacted by mycoviruses, such as *Cryphonectria nitschkei BS122* infected with *C. nitschkei chrysovirus 1,* that show reduction on mycelial growth [91,92]. In *Trichoderma harzianum*, a new species of Partitivirus called Trichoderma harzianum partitivirus 2 (ThPV2) was identified. *T. harzianum* isolates infected by ThPV2 showed higher mycelial density and spore production, which would help virus dispersion in the environment. Furthermore, isolates of *T. harzianum* infected with ThPV2 inhibited many fungal pathogenic species in confrontation tests [93]. These results suggest that ThPV2 can be used as a biological control agent.

### 3.5. The Impact of High-Throughput Sequencing in Mycovirus Studies

The identification/detection of mycoviral species has been occurring for over 50 years. However, the discovery of novel mycoviruses has sharply increased in the last eight years, comprising reports that together account for ~67% of all species described to date (Figure 5A). Indeed, only in 2021, the increase in the number of identified mycoviral species was 25%, with more than 50 new mycoviruses being described. If we consider the diversity of other viral groups that can exceed a thousand viruses already described, those results suggest the mycoviruses’ group is still understudied [94]. This scenario likely reflects the circumstances in which mycoviral infections occur; they usually appear as being cryptic and show little or no impact on the fungal phenotypes, thereby hampering their identification [65]. Furthermore, although a considerable number of mycoviruses have already been described, it should be noted that identification of mycoviral sequences is challenging, since detection and characterization techniques used in phyto- and mycovirology generally comprises electron microscopy techniques, which do not allow the detection of viral molecules without capsid [95]. The recent advances in the identification of mycoviruses have been led by massive parallel sequencing, genomics, and transcriptome studies on various fungal collections [96].

The emergence of HTS platforms have contributed considerably to the identification and characterization of new species and families of mycovirus, as well as to further analyses of their evolutionary history (Figure 5B). Several methods have been applied to environmental samples to identify viral genomes [97,98,99]. For instance, the sequencing of fragmented initiator-linked dsRNA (FLDS) is a relatively new method of a metagenomic approach that allows one to obtain high rates of viral RNA sequences recovery. Another relevant method is RNA sequencing (RNA-seq), a popular method in RNA virus metagenomics that also allows the identification of DNA viruses [100]. Studies suggest that these approaches are sensitive to low levels of viral infection since they present sufficient depth of sequencing; they are also capable to confirm that viral genes are expressed in fungal cells. Over the years, the cost of HTS has decreased and the availability of sequencing data in databases has sharply increased, thus representing an opportunity to exploit existing data and characterize new viruses through bioinformatic approaches [65]. A clear example is an investigation of fungal species of the subphylum *Pezizomycotina,* in which public data were used and 52 novel mycoviral species were identified [65]. It is important to highlight that only the identification of virus species based on HTS does not allow the assessment of genetic, physiological, and ecological aspects of the interactions among viruses and their fungal host species [94]. Nevertheless, such an in silico approach can be viewed as a first-tier screening procedure that can provide support for further studies on biotechnological applications of mycoviruses in agriculture and other akin fields. We expect that with more studies focusing on descriptions of fungal viruses using HTS methods, the picture of the origins and evolution of these viruses become clearer, though more complex depending on the case [27].

### 3.6. Outlook and Perspectives in Mycoviral Research

The asymptomatic nature of mycoviruses has made their study and understanding very challenging [62]. Early studies focused on finding their economic impact as pathogens of yeasts and edible fungi, such as *A. bisporus*, which have presented problems in cell growth [11,72,94,101,102]. Today, the primary motivations for exploring mycoviruses identification is their potential as biological control agents of fungal diseases, which is based mostly on the hypovirulence they impose on their host fungus following virus infection [103]. Considering the examples available of mycoviral species capable of such a virulence attenuation in fungal phytopathogens, these viruses can be valuable tools to develop protection strategies in plants of commercial interest [27]. The first approach to control a fungal disease in seedlings, based on mycovirus, was undertaken in the 1980s, when *C. parasitica* fungal spores containing hypovirulence-causing virus were artificially introduced into fungal populations to control chestnut blight. These results have suggested the use of mycoviruses in biological control; however, this approach overall requires the integration of several factors and components to be effective. Another challenge is the genetic systems and transinfection methods developed for various mycoviruses. The fact that they lack extracellular routes for their transmission makes it difficult to fulfill Koch’s postulates to ensure that the given mycoviruses are indeed the infectious agent that causes hypovirulence in the specific host fungus. An example in which Koch’s postulates were verified was a study with *CHV1* and *C. parasitica,* where infectious cDNA clones were developed and artificially introduced in *C. parasitica* isolates [104]. The study proposes two phases in the process: in the first, after integration into the fungal genome, *CHV1* cDNA clones, are transcribed, up-regulating *C. parasitica* sequences and promoting synthesis of the virus coding strands. In the second phase, *C. parasitica* protoplasts were transformed in vitro with the transcribed synthetic copy of the RNA coding strand [105]. On the other hand, the transmission of mycoviral particles by natural means largely depends on the fungal host’s genetics, as a consequence of the high diversity of vegetative compatibility groups (VCGs) [32,106]. The results obtained on *CHV1*, *Heterobasidium partitivirus 2, 3,* and *13* have shown that mycovirus transmission rates were higher on natural substrates and in the field than in vitro and among isolates of the same species. These results indicate that environmental factors are important variables influencing mycovirus transmission [103].

Some examples of practical biocontrol cases are worth highlighting. Mycovirus-related research has focused on viruses belonging to *Hypoviridae*, mainly because this family includes the above mentioned hypovirus CHV1, a very significant representative that induces hypovirulence in the chestnut blight fungus *Cryphonectria parasitica* [84]. CHV1 is known as the first case of mycoviruses used as biological control agents in Europe, with the *Cryphonectria hypovirus 1* being the best known commercially available control agent so far. Viruses belonging to the *Partitiviridae* have also been extensively studied for their control potential on fungi of the *Heterobasidion* genus; several studies have tried to explain their long-term efficacy, evolution, gene silencing, and genes associated with vegetative incompatibility between fungal hyphae and co-infection between incompatible and geographically unrelated fungi. The *Heterobasidion partivirus 1, 2, 3, 7, 12, 13, 15,* and *P* have been used in the control of *H. annosum, H. paviporum*, and *H. eccrustosum* species infecting European and North American plant species.

It is suggested that understanding mycoviral diversity and methods of identification of new viruses can lead to the discovery of species of interest for different fungal pathogens, or to relevant knowledge for the management of these fungi-derived plant diseases. Based on this view, our group has recently directed research-focus on the possibility of applying mycoviruses in the biocontrol of important cacao diseases. For example, witches’ broom and moniliasis are both caused by fungi of the *Moniliophthora* genus: *M. perniciosa* and *M. roreri*, respectively. *Theobroma cacao L.* is one of the most important crops for an environmentally sustainable agricultural economy in tropical regions of Latin America [107]. These pathogens are generally hard to be controlled, which is achieved by fungicides, cultural practices (such as the pruning of infected parts), and biological control methods, which include the use of other fungal species, such as those of the genus *Trichoderma* [107,108,109,110]. It has been recently shown that the *Trichoderma* species can be affected by *Trichoderma harzianum bipartite mycovirus 1 (ThMV1)*, with interesting and promising results being reported; the presence of this virus has not only improved the biological control capacity of *T. harzianum* against *Fusarium oxysporum*, but has also promoted the growth of cucumber plants used in the experiments [111].

In summary, this integrative review has gathered information on important aspects of mycoviral diversity, which to date has been described in 29 viral families classified by the ICTV and with a total of 262 mycovirus species identified in the databases. Aspects of the biology, taxonomy, and potential biotechnological application of these viral particles were emphasized. Our results showed that HTS and bioinformatics approaches have contributed to the identification of mycoviral species, with a very high number of recent mycoviruses descriptions achieved only in the last two years. Therefore, it is expected that studies on the interactive virus-fungus-plant-environment relationships, as well as further identification of novel mycoviruses, will progressively increase in the coming years. We suggested that the wealth of publicly available information on viral sequences and bioinformatics can be used as first-tier screening strategies for the development of biological control strategies in agricultural systems not yet addressed by the research. Two strategies can be foreseen as worthwhile to pursue: in one, the recognized hypovirulence effects of mycoviruses on their fungal hosts can be applied to relevant plant pathogens, aiming at decreasing their growth and pathogenicity; in the other, hypervirulence interactions can be attempted in fungi known as being beneficial to plants as biocontrol agents, growth promoters, and stress-resistance inducers (e.g., *Trichoderma* spp.). As shown in this review, there is a great deal of possibilities and alternatives of mycoviruses as potential biocontrol agents in many fungi-plant systems, thereby suggesting a promising research area in the near future.

## Figures and Tables

**Figure 1 jof-09-00361-f001:**
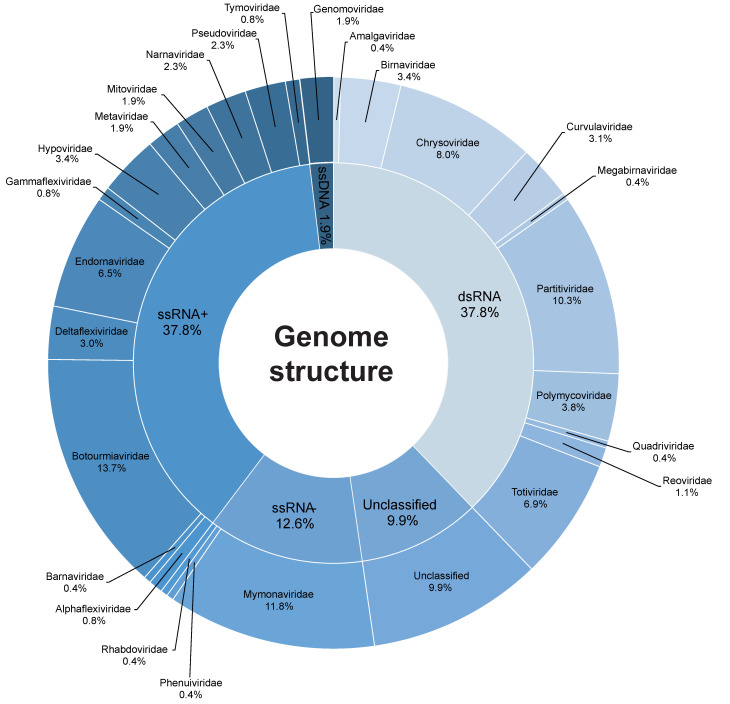
Genomic characteristics of mycovirus species classified by genomic structure.

**Figure 2 jof-09-00361-f002:**
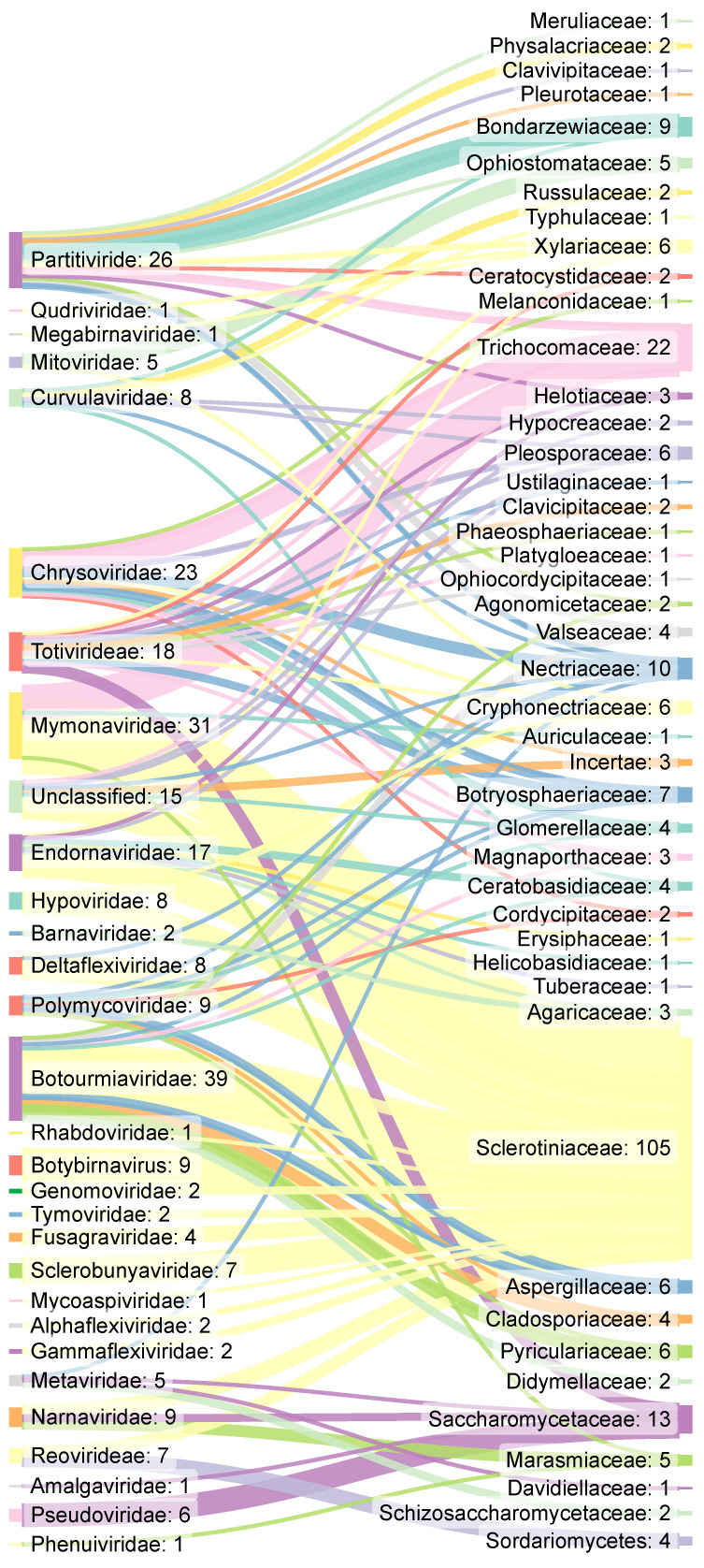
Overview of mycoviruses and fungi host families. The left column represents families of viruses that have been described as infecting fungi while the right column indicates fungal families affected by these mycoviruses. The information flow is represented by the thickness of the lines connecting the columns.

**Figure 3 jof-09-00361-f003:**
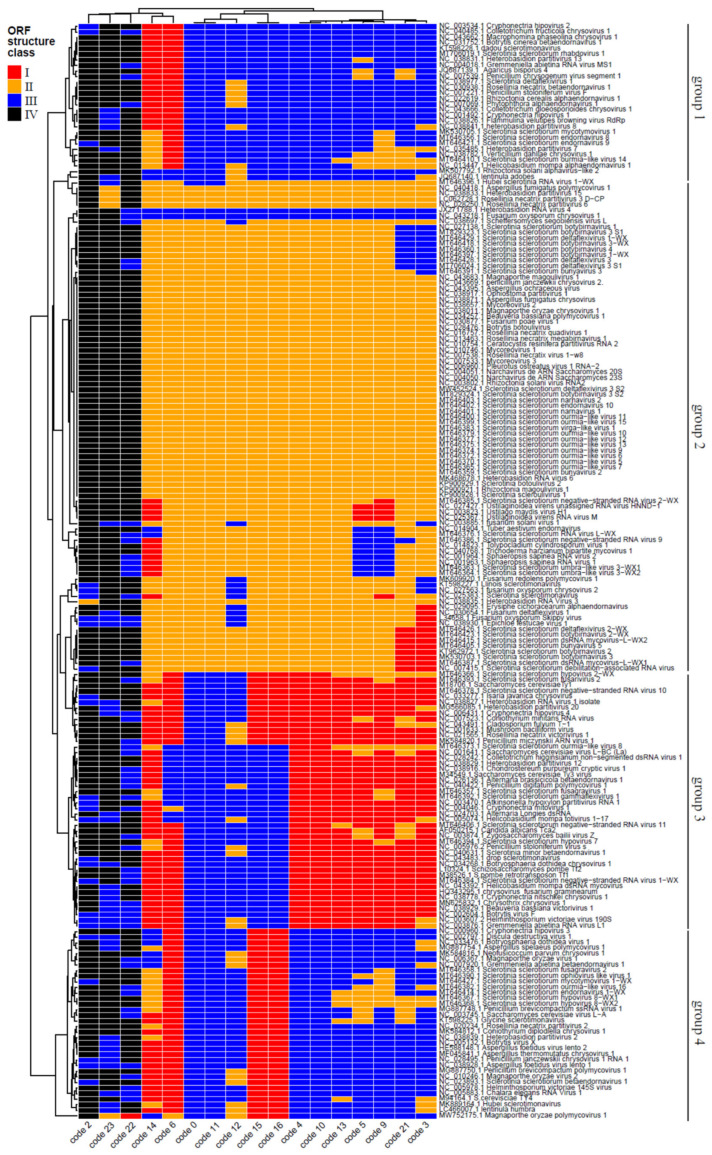
Hierarchical clustering of mycoviral sequences based on the profile of genetic code usage. The ORF structure class legend indicates: I = If the size of the ORF generated by the genetic code is greater than or equal to the size predicted by the standard (1) genetic code, and shows conserved domain; II = If the size of the ORF generated by the genetic code is greater than or equal to the size predicted by the standard (1) genetic code, and lack conserved domain; III = If the size of the ORF generated by the genetic code is smaller than or equal to the size predicted by the standard (1) genetic code, and show a conserved domain; IV = If the size of the ORF generated by the genetic code is smaller than the size predicted by the standard (1) genetic code, and lack conserved domain.

**Figure 4 jof-09-00361-f004:**
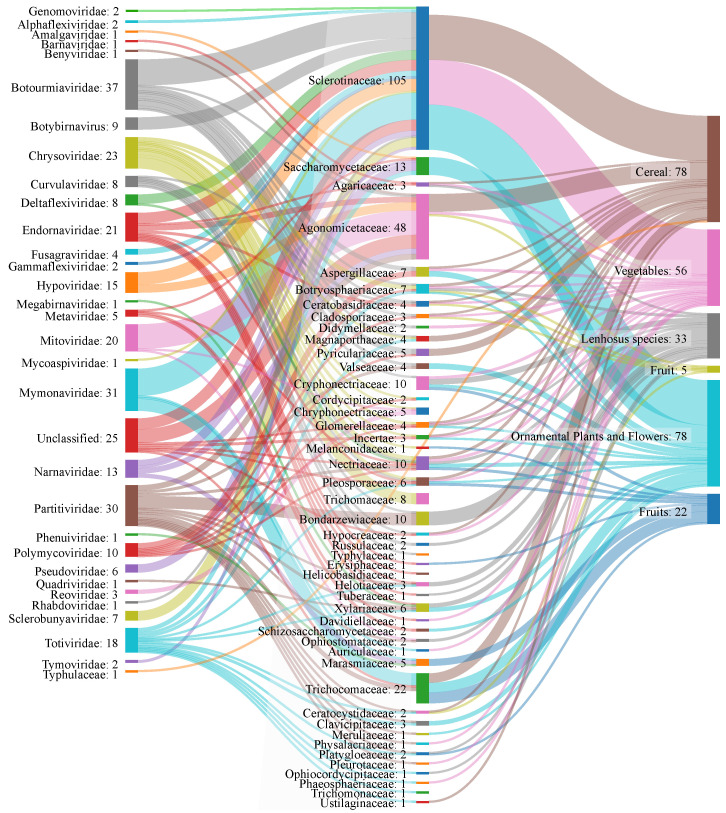
Outlook of the relationship between mycoviruses, fungi, and their plant hosts. The left column represents families of viruses that have been described as infecting fungi; the middle column indicates fungal families affected by these mycoviruses; the right column shows plant classes that are colonized by the fungal families. The information flow is represented by the thickness of the lines connecting the entries in the three columns.

**Figure 5 jof-09-00361-f005:**
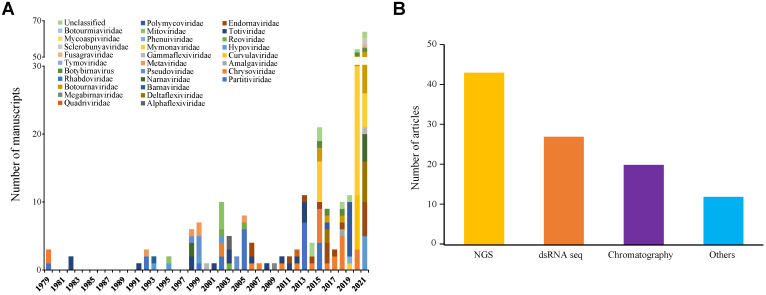
Overview of mycoviruses identification. (**A**) Abundance and family of viruses identified by year. (**B**) Strategy applied to mycovirus identification.

## Data Availability

Not applicable.

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
