# Peer review of "Exploring the Mycovirus Universe: Identification, Diversity, and Biotechnological Applications"

_jof, 2023, doi:10.3390/jof9030361_

Round 1

Reviewer 1 Report

The review carried out on the current state of knowledge and studies realized with Mycoviruses was extensive and very well done, revealing a significant increase in the number of new species accepted by the ICTV and others that may also become new species.The increase in species can be evaluated by consulting the review carried out by Picarelli et al., 2017. In our review there were 96 species accepted by ICTV, 75 dsRNA species and 21 ssRNA species. Most were unclassified species. I believe that the interest in studies with mycoviruses should increase, as they represent a possibility for the biological control of phytopathogenic fungi that cause great damage to the main crops, vegetables, ornamentals, fruit and cereal.

In the article published by Picarelli et al., 2019, we studied the possibility of controlling rhizoctoniosis in grasses by mycovirus and we identified 47 different mycoviruses. In our study we reported sequences corresponding to mostly new viral species belonging to the positive sense ssRNA genome virus groups (Mitovirus, Botourmiaviridae, Hypovirus, Endornavirus, Hepe-Virga-like), and to dsRNA virus groups such as the Alphapartitivirus and Gammapartitivirus, in Rhyzoctonia solani. The mitoviruses RsMV24 and RsMV29, and the endornavirus RsEV7, were the most prevalent viruses since they occurred in all the isolates. Although none of the isolates under scrutiny were hypovirulent, such a library of mycoviruses could be the basis for a targeted virus-induced gene silencing (VIGS) approach. 

With regard to the current revision I have few suggestions, but in the figure 4: I suggest the incorporation of the presence of mitovirus in R. solani that occurs in grasses in the family Ceratobasidiaceae and it would be better to change the term cereal to poaceae to include grasses. I also suggest the incorporation of the mycoviruses identified in the article by Picarelli et al., 2019, since there is no information on the identification of mycoviruses in R. solani and there are several cases described, including in grass in the mentioned article.

Author Response

Reviewer 1 The review carried out on the current state of knowledge and studies realized with Mycoviruses was extensive and very well done, revealing a significant increase in the number of new species accepted by the ICTV and others that may also become new species.The increase in species can be evaluated by consulting the review carried out by Picarelli et al., 2017. In our review there were 96 species accepted by ICTV, 75 dsRNA species and 21 ssRNA species. Most were unclassified species. I believe that the interest in studies with mycoviruses should increase, as they represent a possibility for the biological control of phytopathogenic fungi that cause great damage to the main crops, vegetables, ornamentals, fruit and cereal. In the article published by Picarelli et al., 2019, we studied the possibility of controlling rhizoctoniosis in grasses by mycovirus and we identified 47 different mycoviruses. In our study we reported sequences corresponding to mostly new viral species belonging to the positive sense ssRNA genome virus groups (Mitovirus, Botourmiaviridae, Hypovirus, Endornavirus, Hepe-Virga-like), and to dsRNA virus groups such as the Alphapartitivirus and Gammapartitivirus, in Rhyzoctonia solani. The mitoviruses RsMV24 and RsMV29, and the endornavirus RsEV7, were the most prevalent viruses since they occurred in all the isolates. Although none of the isolates under scrutiny were hypovirulent, such a library of mycoviruses could be the basis for a targeted virus-induced gene silencing (VIGS) approach. 1. With regard to the current revision I have few suggestions, but in the figure 4: I suggest the incorporation of the presence of mitovirus in R. solani that occurs in grasses in the family Ceratobasidiaceae and it would be better to change the term cereal to poaceae to include grasses. I also suggest the incorporation of the mycoviruses identified in the article by Picarelli et al., 2019, since there is no information on the identification of mycoviruses in R. solani and there are several cases described, including in grass in the mentioned article. Answer. We appreciate the reviewer suggestion. In this revised version of our manuscript, we added in Figure 4 and Supplementary Table 4 the information of mycoviruses described by the study of Picarelli et al., 2019.

Reviewer 2 Report

The manuscript of “Exploring the mycovirus universe: Identification, Diversity, and Biotechnological Applications” gives the good review on the identification, diversity, and biotechnological applications though screening all of the data of published mycovirus species, which is important for the conclusion of the mycovirus researches. However, there are also many places need to be revised and improved. If the places have been revised, the manuscript could be accepted.

1.     In the whole research, a big part analysis is on the genetic code usage, however, this should not for the replication strategy, but for the translation strategy for protein. 

2.     On the line 95, The “managing” should be changed into “manage”.

3.     On the line 142, “Articles referring to applications of fungi or viruses in human diseases, perspectives 140 of applications of viruses in human diseases, or as cause variables in the development of 141 cancer and old articles published prior to the year 2000, were excluded due to the updating of biological concepts on mycoviruses research”, so here the reference for the “biological concepts on mycoviruses research” are needed.

4.     On the line 222, “the clustering analysis based on four ORFs structures” and the “17 distinct genetic codes” are all need to be stated more clearly, in the supplementary, some mycoviruses only have two ORFs. What are the four ORFs for the mycovirus? and the origin and development about the 17 distinct genetic codes need to be described clearly in the background.

5.     In the researches of the mycovirus involving the genetic codes, could you added the accurate report for the differently distinct genetic codes using?

6.     Could all of involved mycoviruses with the distinct genetic codes be elaborated in the background? And if all of involved mycovirus are really reported using the distinct genetic codes? Which could help you trigger the problem and then you could give the answer in the result part. 

7.     On the line 232, could you describe “the conserved domain” in the background? which need give the elaboration, and in the supplementary materials, also need to be added. 

8.     On the line 485, if the “[130,132]” need changed into “[130-132]”?

9.     On the line 489, the sentence is not for “[133]”, you need check if the number of reference is right.

10.  Moreover, about the biological control function of the mycovirus, there is a new mycovirus published in 2022, which not only could improve the biocontrol function of the host, but also could promote onset of flowering: Wang R, Liu C, Jiang X, Tan Z, Li H, Xu S, Zhang S, Shang Q, Deising HB, Behrens, SE, Wu B*, The Newly Identified Trichoderma harzianum Partitivirus (ThPV2) Does Not Diminish Spore Production and Biocontrol Activity of Its Host. Viruses 2022, 14, 1532. https://doi.org/10.3390/v14071532.

Author Response

The manuscript of “Exploring the mycovirus universe: Identification, Diversity, and Biotechnological Applications” gives the good review on the identification, diversity, and biotechnological applications though screening all of the data of published mycovirus species, which is important for the conclusion of the mycovirus researches. However, there are also many places need to be revised and improved. If the places have been revised, the manuscript could be accepted.

  1. In the whole research, a big part analysis is on the genetic code usage, however, this should not for the replication strategy, but for the translation strategy for protein. 

Answer. We agree with the reviewer. We changed our text to reflect the referee comment regarding the genetic code usage is related with the translation strategy.

  1. On the line 95, The “managing” should be changed into “manage”.

Answer. We thank the reviewer for the suggestion. We have changed the text as suggested.

  1. On the line 142, “Articles referring to applications of fungi or viruses in human diseases, perspectives 140 of applications of viruses in human diseases, or as cause variables in the development of 141 cancer and old articles published prior to the year 2000, were excluded due to the updating of biological concepts on mycoviruses research”, so here the reference for the “biological concepts on mycoviruses research” are needed.

Answer. We thank the reviewer for the suggestion. We have moved the sentence for a best fit paragraph and added the reference as requested (line 114-117).

  1. On the line 222, “the clustering analysis based on four ORFs structures” and the “17 distinct genetic codes” are all need to be stated more clearly, in the supplementary, some mycoviruses only have two ORFs. What are the four ORFs for the mycovirus? and the origin and development about the 17 distinct genetic codes need to be described clearly in the background.

Answer. We apologize to the referee for the lack of clarity. We have updated the Figure 3 and the text to improve the comprehension that the ORF structure is regarding the length of the ORF and the presence of conserved domains in this ORF when predicted with different genetic codes. We also indicated in the figure the different classes of ORFs when we tested the genetic codes and the major groups containing viral species with similar behavior regarding genetic code possibilities.

  1. In the researches of the mycovirus involving the genetic codes, could you added the accurate report for the differently distinct genetic codes using?

Answer. We apologize to the referee for the lack of clarity. We have included all the genetic codes tested at the Methodology section (lines 131-139).

  1. Could all of involved mycoviruses with the distinct genetic codes be elaborated in the background? And if all of involved mycovirus are really reported using the distinct genetic codes? Which could help you trigger the problem and then you could give the answer in the result part. 

Answer. We apologize to the referee for the lack of clarity. Most of the mycoviruses described were only described for the first time with the genetic codes Standard (1) or Mitochondrial (4). We have only investigated the possibility of use distinct genetic codes based on the ORF and protein produced when testing those codes. We believe that the final assumption of genetic code usage only could be done by in vitro/in vivo experimental analysis. However, our data could help indicating the “more probably hosts” based on genetic code compatibility.

  1. On the line 232, could you describe “the conserved domain” in the background? which need give the elaboration, and in the supplementary materials, also need to be added.

Answer. We thank the referee for pointing this out and would like to apologize for our mistake. The Supplementary Table 3 cited on the main text included all the information regarding the conserved domains identified in each genome analyzed for each of the genetic codes tested. We have included the file in this revised version of our manuscript.

  1. On the line 485, if the “[130,132]” need changed into “[130-132]”?

Answer. We apologize to the referee for the mistake. We have corrected the reference in this sentence.

  1. On the line 489, the sentence is not for “[133]”, you need check if the number of reference is right.

Answer. We apologize to the referee for the mistake. We have checked the reference and corrected the citation the text.

  1. Moreover, about the biological control function of the mycovirus, there is a new mycovirus published in 2022, which not only could improve the biocontrol function of the host, but also could promote onset of flowering: Wang R, Liu C, Jiang X, Tan Z, Li H, Xu S, Zhang S, Shang Q, Deising HB, Behrens, SE, Wu B*, The Newly Identified Trichoderma harzianum Partitivirus (ThPV2) Does Not Diminish Spore Production and Biocontrol Activity of Its Host. Viruses 2022, 14, 1532. https://doi.org/10.3390/v14071532.

Answer. We appreciate the reviewer's suggestion. We included in the text in the section “Cases of mycoviruses as biological control agents (hypovirulent phenotypes)” about the study carried out by Wang et al. 2022(Lines 440-445).

Reviewer 3 Report

Diana et al have tried to describe “identification, diversity, and biotechnological applications of mycoviruses”. However, for some reasons, I do not recommend it to be published in its current form in this journal.

1.     In Figure 1, the classification of many mycovirus families is wrong. For example, the Mitoviridae belongs to positive RNA viruses instead of dsRNA viruses, the Metaviridae does not belong to ssDNA viruses but to ssRNA viruses. The Hypoviridae and Endornaviridae belong to ssRNA viruses. Please delete the "dsRNA and ssRNA" category and arrange the classification according to the ICTV report.

2.     The analysis of genetic code usage in Figure 3 should be a new data, but the description of the related text and legend is not easy to understand and does not seem to draw a clear and scientific conclusion. In addition, the quality of the map is too low.

3.     The description of lines 135-293 looks disorganized. Why can't it be subtitled as clearly as the content that follows?

Author Response

Diana et al have tried to describe “identification, diversity, and biotechnological applications of mycoviruses”. However, for some reasons, I do not recommend it to be published in its current form in this journal.

  1. In Figure 1, the classification of many mycovirus families is wrong. For example, the Mitoviridae belongs to positive RNA viruses instead of dsRNA viruses, the Metaviridae does not belong to ssDNA viruses but to ssRNA viruses. The Hypoviridae and Endornaviridae belong to ssRNA viruses. Please delete the "dsRNA and ssRNA" category and arrange the classification according to the ICTV report.
  2. A. We apologize to the reviewer. We checked all the data and in this new version, we believe that Figure 1 is correct.

  1. The analysis of genetic code usagein Figure 3 should be a new data, but the description of the related text and legend is not easy to understand and does not seem to draw a clear and scientific conclusion. In addition, the quality of the map is too low.
  2. A. In this new version, we made a new version of Figure 3 and edited this part of the text (Lines 247 -278).

  1. The description of lines 135-293 looks disorganized. Why can't it be subtitled as clearly as the content that follows?
  2. A. We agree with the reviewer. In this new version of our manuscript, we have restructured the text to make it more fluid.

Reviewer 4 Report

The manuscript describes a comprehensive review using Scopus, Web of Science and PubMed databases. Authors explore their molecular features and their use in biotechnology.

The review paper is clearly written and I recommend this MS for publication, after a minor correction of the typos.

There are a few lines with minor errors here.

line 26 and 29 - Sclerotiniaceae

In any future submission, please follow Instruction for authors and follow the style of References!

ref 28 and others-Rhizoctonia Solani should be Rhizoctonia solani.

refs 4, 12, 13, 14, 21, 26, 47-52, 54-56, 58, 73, 75, 78, 101, 107, 109, 117, 121, 125, 132, and 133 should be revised.

Author Response

The manuscript describes a comprehensive review using Scopus, Web of Science and PubMed databases. Authors explore their molecular features and their use in biotechnology.

The review paper is clearly written and I recommend this MS for publication, after a minor correction of the typos.

  1. We thank the reviewer for the suggestions. All changes were made in this new version of the manuscript.

There are a few lines with minor errors here.

line 26 and 29 – Sclerotiniaceae

  1. We changed this word in the new version of our manuscript.

In any future submission, please follow Instruction for authors and follow the style of References!

  1. In this version, we checked the format of all references.

ref 28 and others-Rhizoctonia Solani should be Rhizoctonia solani.

refs 4, 12, 13, 14, 21, 26, 47-52, 54-56, 58, 73, 75, 78, 101, 107, 109, 117, 121, 125, 132, and 133 should be revised.

  1. We italized the species name in the references and checked all the references of our manuscript.

Round 2

Reviewer 2 Report

The manuscript of “Exploring the mycovirus universe: Identification, Diversity, and Biotechnological Applications” gives the good review on the identification, diversity,  the distinct genetic codes, biotechnological applications though screening all of the data of published mycovirus species, which is important for the conclusion of the mycovirus researches, and also gave more value revising. However, in the published paper, I am not paid attention on the  comparing about the distinct genetic codes, so what's the suggestions from you for the researcher on the mycovirus later? do you think they need use different genetic codes to check the protein sequences? 

Author Response

We thank the reviewer for the comment. Indeed, we still do not fully understand the impact of the use of distinct genetic codes on mycoviruses fitness and/or behavior. Most of which is known is derived from the identification of viruses that infect multiple hosts from different kingdoms, such as fungi and plant, showing the possibility of codon usage consistent with both organisms.

However, we can suppose that one of the main impacts of using a different genetic code is that it can allow the virus to produce proteins that are different from those of the host. This can give the virus an advantage in evading the host's immune system and in adapting to new environments. For example, some viruses have been shown to use a different genetic code to produce proteins that are resistant to antiviral drugs.

Another impact of using a different genetic code is that it can affect the efficiency of protein synthesis. Some genetic codes are more efficient than others, meaning that they can produce proteins more quickly and with fewer errors. By using a more efficient genetic code, the virus can produce its proteins more quickly and effectively, giving it a competitive advantage over the host.

Finally, using a different genetic code can also affect the way that the virus interacts with the host cell. For example, some viruses have been shown to use a different genetic code to produce proteins that interact with the host cell's machinery in different ways. This can lead to changes in the way that the virus replicates and spreads within the host, as well as changes in the symptoms and severity of the resulting disease.

In summary, the use of different genetic codes by viruses can have a range of impacts on their host strategy, including changes to protein production, efficiency, and interaction with host cells. Since we believe it is a important discussion, we have include this further discussion on the Discussion Topic of our manuscript (Lines 242 - 260).

Reviewer 3 Report

Thanks to the authors for making appropriate revisions to my suggestions. However, Figure 1 still contains errors in the taxonomy of mycoviruses. So far, some ssDNA mycoviruses have been reported and have been approved by ICTV to enter the family Genomoviridae. So add the "ssDNA" category and the " Genomoviridae family". Also, as far as I know, no dsDNA mycoviruses have been reported, so please delete the "dsDNA and..." Category. Overall, I recommend that the article be accepted with minor modifications.

Author Response

We apologize to the referee for the mistake. Indeed, there are no fungi-infecting dsDNA virus described in the literature as we can observe in the Supplementary Table S1. The presence of dsDNA in the Donut plot was a mistake that has been corrected, since the correct class was ssDNA as pointed out previously.